# Circular Photogalvanic Current in Ni-Doped Cd_3_As_2_ Films Epitaxied on GaAs(111)B Substrate

**DOI:** 10.3390/nano13131979

**Published:** 2023-06-29

**Authors:** Gaoming Liang, Guihao Zhai, Jialin Ma, Hailong Wang, Jianhua Zhao, Xiaoguang Wu, Xinhui Zhang

**Affiliations:** 1State Key Laboratory of Superlattices and Microstructures, Institute of Semiconductors, Chinese Academy of Sciences, Beijing 100083, China; gmliang@semi.ac.cn (G.L.); zhai_guihao@semi.ac.cn (G.Z.); jlma1991@semi.ac.cn (J.M.); allen@semi.ac.cn (H.W.); jhzhao@red.semi.ac.cn (J.Z.); xgwu@red.semi.ac.cn (X.W.); 2Center of Materials Science and Optoelectronics Engineering, University of Chinese Academy of Sciences, Beijing 100049, China

**Keywords:** Ni-doped Dirac semimetal Cd_3_As_2_, the circular photogalvanic effect, the deep-level magnetic impurity bands

## Abstract

Magnetic element doped Cd_3_As_2_ Dirac semimetal has attracted great attention for revealing the novel quantum phenomena and infrared opto-electronic applications. In this work, the circular photogalvanic effect (CPGE) was investigated at various temperatures for the Ni-doped Cd_3_As_2_ films which were grown on GaAs(111)B substrate by molecular beam epitaxy. The CPGE current generation was found to originate from the structural symmetry breaking induced by the lattice strain and magnetic doping in the Ni-doped Cd_3_As_2_ films, similar to that in the undoped ones. However, the CPGE current generated in the Ni-doped Cd_3_As_2_ films was approximately two orders of magnitude smaller than that in the undoped one under the same experimental conditions and exhibited a complex temperature variation. While the CPGE current in the undoped film showed a general increase with rising temperature. The greatly reduced CPGE current generation efficiency and its complex variation with temperature in the Ni-doped Cd_3_As_2_ films was discussed to result from the efficient capture of photo-generated carriers by the deep-level magnetic impurity bands and enhanced momentum relaxation caused by additional strong impurity scattering when magnetic dopants were introduced.

## 1. Introduction

As a representative material of the three-dimensional Dirac semimetals (3D DSMs), Cd_3_As_2_ has been extensively studied owing to its outstanding features of good stability in air, high carrier mobility and Fermi velocity [1,2,3]. Its topological nature has been verified by the angle-resolved photoemission spectroscopy (ARPES) and quantum transport experiments [2,3,4,5,6]. The giant linear magnetoresistance [1,7,8], Landau levels [5,9], quantum Hall effect [10,11], superconductivity [12,13] and magneto-optical phenomena [14], etc., have been studied to reveal its novel physical properties. Cd_3_As_2_ is also proposed to be suitable for high-speed, broadband photodetectors, or mid-infrared optical switches and modulators due to its advantages of having high carrier mobility and linear dispersion of band energy [15,16,17,18,19,20].

Circular photogalvanic effect (CPGE) can generate a spin-polarized charge current by creating an asymmetric carrier population in momentum space [21,22], under circularly-polarized light excitation with the angular momentum selection rule. CPGE has served as an excellent experimental method to study the spin-associated properties in low-dimensional semiconductors [23,24,25,26], topological insulators (TIs) [27,28,29,30,31], Weyl semimetals [32,33,34,35,36,37,38,39], and 2D Dirac material graphene [40,41,42,43,44]. In theory, the CPGE was predicted to be zero in an ideal Dirac system because of the symmetric photoexcitation with respect to the Dirac point [39,45,46], as a consequence of both the time-reversal and spatial inversion symmetry in DSMs. However, under symmetry breaking caused by the epitaxial strain or Schottky electric field, CPGE has been observed experimentally in Cd_3_As_2_ films and Cd_3_As_2_ Nanobelts [47,48]. It is revealed that symmetry breaking in DSMs can be induced by introducing strain, dimensional control, or doping magnetic elements [5,49,50,51]. The electronic and magnetic properties of 3d transition metal element (3d-TM) doped Cd_3_As_2_ have been theoretically investigated by using first principle calculation with density functional theory. Cr-doped Cd_3_As_2_ was proposed to be a promising room-temperature ferromagnetic semiconductor via Monte Carlo simulations [52]. It is reported that 3d-TM doping reduces the symmetry of Cd_3_As_2_ in the Cr- and Mn-doped Cd_3_As_2_, leading to a band gap opening [15,16,52,53,54,55]. Meanwhile, magnetic doping was found to make a great difference in the relaxation time of photoexcited carriers by ultrafast spectral experiments in the Cr- and Mn-doped Cd_3_As_2_ [15,56]. In addition, Mn doping was reported to influence the transport properties of Cd_3_As_2_ thin films grown on GaAs(111)B substrate [57]. Given that magnetic doping could cause symmetry breaking and possible phase transition from DSM to Weyl semimetal for Cd_3_As_2_, it is highly desirable to explore CPGE photocurrent in magnetic element doped Cd_3_As_2_, to further explore its spin-related physical properties and potential opto-electronic applications. However, there is no CPGE study available yet in magnetically doped Cd_3_As_2_ so far.

In this work, we systematically measured the CPGE current generation for the Ni-doped Cd_3_As_2_ films with different doping concentrations and compared it with that in the undoped one at various temperatures. The CPGE current can be observed in all Cd_3_As_2_ films and shows a general increase with a larger optical incident angle and pumping power. The CPGE current measured in the Ni-doped Cd_3_As_2_ films was greatly reduced compared with that from an undoped one under the same experimental conditions. The CPGE current generated in the Ni-doped Cd_3_As_2_ films was suggested to originate from the reduced structural symmetry by the large epitaxial strain and magnetic doping, rather than a possible phase transition to Weyl semimetal by magnetic doping. Moreover, a complex temperature variation of CPGE current was observed in the Ni-doped Cd_3_As_2_ films, which was attributed to the efficient capture of photo-generated carriers upon optical excitation by the deep-level magnetic impurity bands and the enhanced momentum relaxation with magnetic element doping.

## 2. Materials and Methods

The experiment was carried out on the 20 nm-thick Ni-doped Cd_3_As_2_ films with different doping concentrations and the undoped control sample. All films were grown on GaAs(111)B substrate by molecular beam epitaxy at a low temperature of 180 °C. During the growth process, the molecular beams of Cd, Ni and As were controlled by using three isolated effusion cells. For the Ni-doped Cd_3_As_2_ films with different doping concentrations, the beam equivalent pressure ratio of Ni to Cd was set to be 2%, 4% and 8%, and the corresponding Ni source growth temperature was 1130 °C, 1170 °C and 1210 °C, respectively. The nominal Ni concentration was 2%, 4% and 8% and denoted as sample-A (Ni-2%), sample-B (Ni-4%) and sample-C (Ni-8%), respectively. Since the epitaxial Cd_3_As_2_ thin films would be easily oxidized if exposed to the air, and this can cause an adverse effect of electrical and optical properties, a 3-nm-thick GaAs capping layer was grown at the same low temperature of 180 °C to prevent the film from oxidation. The Curie temperature of the Ni-doped Cd_3_As_2_ films is about 45 K. More detailed sample growth procedures and characterization results can be found in our previous studies [47,58]. The experimental setup is shown schematically in Figure 1: the *x*-*y* coordinates plane is defined as being parallel to the sample’s surface and the *z*-axis is perpendicular to the sample’s surface. We made a pair of ohmic electrodes by indium deposition along the *x*-axis with a 3 mm distance far apart. A Ti: sapphire laser (Chameleon Ultra II, Coherent Inc., Santa Clara, CA, USA) with a repetition rate of 80 MHz, was utilized as an exciting light source in our experiment. The oblique incidence plane of the laser is perpendicular to the *x*-axis. The irradiated laser spot is tuned to locate in between the two electrodes with a radius of about 1 mm (the cleaved sample is about 5 × 5 mm in size). We used a polarizer and a λ/4 wave-plate to alter the light helicity *P*c = sin 2*φ* from the left-handed (σ^−^, *P*c = −1) to right-handed (σ^+^, *P*c = 1), here *φ* is the angle between the polarization direction of the polarizer and the optical axis of the λ/4 plate. Then the photocurrent is measured by a preamplifier and a lock-in amplifier that is in phase with an optical chopper working at a fixed frequency. All samples were placed in a Janis closed-cycle optical cryostat for low-temperature measurements.

## 3. Results and Discussion

Figure 2a,b shows the representative photogalvanic current response by varying angle *φ* measured at 10 K and an incident angle of 45°, under excitation of 750 nm and 900 nm with pumping power of 14 mW for the Ni-4% doped sample-B. Though Cd_3_As_2_ thin films were epitaxied on GaAs substrate and capped with a 3-nm-thick GaAs which was also grown at a low temperature of 180 °C (LT-GaAs) layer, as depicted in the sample preparation part, the possible photogalvanic current contribution from either GaAs buffer/substrate or GaAs capping layers can be ruled out, by considering the facts of nearly amorphous LT-GaAs capping layer and the control study results of a bare GaAs buffer sample, as already discussed in our previous work [47]. In addition, our photogalvanic current study on Ni-4% samples with different excitation wavelengths further confirms this and will be discussed later. The photogalvanic current includes two components with different periodicities and is usually written as [33]
(1)$$j_{\lambda }=j_{C}\sin\left(2\varphi +\psi_{C}\right)+j_{L}\sin\left(4\varphi +\psi_{L}\right)+j_{0},$$

Here, *j_C_* and *j_L_* are the amplitude of CPGE and linear photogalvanic effect (LPGE) current, respectively. *ψ_C_* (*ψ_L_*) represents the initial phase of CPGE (LPGE). *j*_0_ is the background current resulting from the photovoltaic effect at electrodes or the Dember effect and does not rely on the helicity of the excitation laser [21]. By fitting the measured photogalvanic current with Equation (1), the CPGE (LPGE) current response can be separately determined and was shown by the blue (red) dashed lines in Figure 2. It was observed that the extracted CPGE current in the Ni-4% sample at the radiation wavelength of 900 nm and 750 nm is 1.4 nA and 1.5 nA, respectively. This finding further suggests that the measured CPGE photocurrent should be generated mainly in the Ni-doped Cd_3_As_2_ film itself, rather than the upper GaAs capping layer or the GaAs buffer/substrate. Since the photon energy of radiation wavelength at 900 nm is lower than that of GaAs bandgap, thus we can exclude the possible CPGE generation from GaAs capping/buffer layer induced by interband excitation. In addition, the generated CPGE with radiation wavelengths of 900 nm and 750 nm is almost consistent within the experimental error, which further rules out the possible intraband transition associated with CPGE generation in GaAs capping/buffer layer.

Figure 3a shows the extracted CPGE amplitude by fitting with Equation (1) at various incident angles *θ* in the Ni-4% sample, measured with the excitation power of 21 mW at 900 nm. One can see that, the CPGE current increases with the incident angle within the measured angle range from 0 to 60 degrees, this is consistent with the phenomenological theory of CPGE described by the following formula [21]:(2)$$j_{\lambda }=\gamma_{\lambda \mu }i{\left(E\times E^{*}\right)}_{\mu },$$

Here, *j_λ_* represents the generated CPGE photocurrent density, *γ_λμ_* represents the second-rank pseudo-tensor associated with the symmetry of the material, *λ* goes through all the three Cartesian coordinates *x*, *y*, *z*, *μ* = *x*, *y*, *z*, **E** represents the complex amplitude of the electric field of the incident excitation light, its unit vector $\hat{e}$ points to the direction of light propagation. $\gamma_{\lambda \mu }i{\left(E\times E^{*}\right)}_{\mu }$ can be transformed into $\gamma P_{circ}E^{2}{\hat{e}}_{\left|\right|}$, where $P_{circ}$ denotes the circular polarization degree. That is, the CPGE current is proportional to the field projection of the incident light onto the sample surface ($\propto E^{2}{\hat{e}}_{\left|\right|}$), as well as the radiation intensity. Figure 3b shows the measured CPGE current at 10 K in sample B with various photoexcitation power excited at 900 nm and an incident angle of 45°. It is seen that the CPGE increases nearly linearly with photoexcitation power from 3 to 27 mW, which agrees with the theoretical expectation.

To study the effect of magnetic doping on CPGE current generation, we further measured the helicity-dependent photocurrent responses for the Ni-doped Cd_3_As_2_ films with different doping concentrations and compared the results with an undoped control sample. The extracted CPGE current by fitting the original data taken at room temperature with Equation (1), normalized with the corresponding photoexcitation power, is shown in Figure 4. The inset shows the zoom-in CPGE results for Cd_3_As_2_ films with different Ni dopant concentrations. We can see that the measured CPGE current in the undoped 20 nm Cd_3_As_2_ is two orders of magnitude larger than that in the Ni-doped Cd_3_As_2_ films. For the Ni-doped Cd_3_As_2_ thin films, the magnetic element doping can introduce time inversion symmetry breaking as theoretically expected, leading to possible phase transition from Dirac to Weyl semimetals, so that stronger CPGE current than the undoped ones might be expected. However, our experiment shows that the magnetic element doping did not enhance the CPGE current generation, on the contrary, the generated CPGE current in the Ni-doped Cd_3_As_2_ films is much smaller than that in the undoped one, as shown in Figure 4. Therefore, we believe that the CPGE current generation in the Ni-doped Cd_3_As_2_ films also originates from the structural symmetry breaking induced by lattice strain and Ni doping, similar to that in the undoped Cd_3_As_2_ thin films discussed in our earlier study [47]. It would be hard to give a quantitative estimation of the additional strain introduced by 2%, 4%, and 8% Ni doping, though, the dominant strain in the Ni-doped Cd_3_As_2_ films is believed to result from the large lattice mismatch of 10% between Cd_3_As_2_ and GaAs(111)B substrate, the same as the undoped ones. To analyze the observed difference in the CPGE current generation efficiency in the doped and undoped films, we first compared the parameters (Electron density and Hall mobility) closely related to the CPGE current response, thanks to the characterization results of our samples in previous work [58]. Table 1 lists the electron density and Hall mobility at 3 K for the Ni-doped Cd_3_As_2_ films with different doping concentrations and the undoped ones. It is seen that the electron density and Hall mobility of the Ni-doped Cd_3_As_2_ films with different doping concentration and the undoped one is almost on the same order of magnitude, which cannot account for the huge difference in CPGE current generation as shown in Figure 4. We thus further compare the photo-excited carrier generation and its relaxation dynamics for the doped and undoped films, since CPGE current is also closely associated with the momentum relaxation process and carrier lifetime.

Figure 5 shows typical transient reflection spectra (Δ*R*/*R*) of different samples, measured with pump and probe wavelengths of 800 nm and 3500 nm at 4.5 K, and more transient dynamics results can be found in our previous work [59]. Apparently, the photo-excited carrier decay time of the Ni-doped films, take the result of sample-B as an example, is only ~0.75 ps, much shorter than the undoped film (~3.98 ps). This shorter photo-excited carrier lifetime in the Ni-doped films results from the greatly enhanced electron-phonon and electron-impurity scattering introduced by Ni doping [56,59]. Meanwhile, the absolute peak value of Δ*R*/*R* for the undoped sample was found to be about three times of the doped ones. The peak value of Δ*R*/*R* is closely relevant to the optically excited carrier concentration according to previous work [60]. As a result, compared with the undoped Cd_3_As_2_ film, the Ni-doped samples have a shorter carrier lifetime and lower photo-excited carrier density under the same excitation condition. In addition, our previous study also showed that there appeared a long decay time of tens of picoseconds at the elevated temperature or higher optical pumping power, for all Ni-doped films [59]. This long carrier decay process has also been reported in Mn-doped Cd_3_As_2_ and was associated with the deep-level magnetic impurity level near the Cd_3_As_2_ Fermi surface [56]. The rather weak temperature variation of this long decay process is a characteristic of the trapped carriers by the magnetic impurity states in Mn- or Ni-doped Cd_3_As_2_ films [56,59]. Our transient optical reflection studies for the measured films, together with their electrical characterization, suggest that the efficient capture of photo-generated carriers by the deep-level magnetic impurity bands, and the greatly enhanced momentum relaxation caused by stronger electron-phonon and electron-impurity scattering introduced by Ni doping, effectively reduce the CPGE current generation in the Ni-doped Cd_3_As_2_ films compared with the undoped one.

Furthermore, we made a comparison investigation on the helicity-dependent photocurrent for the undoped Cd_3_As_2_ film and sample-B (Ni-4%) at various temperatures from 10 to 300 K. Figure 6a shows the extracted CPGE current as a function of temperature in sample-B excited at 900 nm and the undoped Cd_3_As_2_ film excited at 750 nm, here the generated CPGE current is normalized by the optical excitation power. It is seen that the CPGE current generation efficiency in the undoped Cd_3_As_2_ film shows a general increase with rising temperature. However, for the Ni-4% doped sample, the CPGE current generation efficiency shows a complex temperature variation: it first gradually increases when increasing temperature from 10 to 75 K, but then decreases in the temperature range of 75–200 K, and eventually increases again with rising temperature from 200 to 300 K. Since the photon energy at both excitation wavelength of 750 nm and 900 nm is much higher than the linear Dirac band energy range of Cd_3_As_2_, in addition, the CPGE current generated with excitation wavelength of 750 nm and 900 nm is almost the same under the same experimental conditions as shown in Figure 2, so the different temperature variation of the CPGE current generation efficiency excited at 750 nm and 900 nm for the undoped and Ni-doped Cd_3_As_2_ films, should be mainly caused by Ni doping, rather than the electron band structure. Therefore, we further discuss the effect of Ni doping on CPGE current in Cd_3_As_2_ films. According to previous works [61,62], the CPGE current can be described by
(3)$$J_{CPGE}=e\alpha_{e}G\tau_{p}P_{c}S\mu ,$$

Here, *α_e_* represents the effective electric field caused by the spin-orbit coupling (SOC), *e* is the elementary charge, *G* is the generation rate of the spin-polarized electrons, *τ_p_* and *μ* are the momentum relaxation time and mobility of the electrons, *P_c_* is the circular polarization degree, *S* is the cross-section area of the current. Under an applied DC bias, the photoconductivity current *I*_0_ can be written as [61,62]
(4)$$I_{0}=eEG\tau_{0}S\mu ,$$

Here, *E* is the applied electric field and *τ_0_* is the recombination lifetime of the photo-excited carriers. To eliminate the influence of the optical absorptivity and the electron mobility at various temperatures, one can normalize the CPGE current with respect to *I*_0_ as
(5)$$\frac{J_{CPGE}}{I_{0}}=\frac{\alpha_{e}\tau_{p}}{E\tau_{0}},$$

In our experiment, we measured the temperature-dependent resistance of the Ni-4% sample as shown in the inset of Figure 6a. One can see that the resistance first increases with temperature (10–150 K) and then decreases within a temperature range of 150–300 K, indicating that the Ni-doped Cd_3_As_2_ film undergoes a phase transition from semimetal to semiconductor (a trivial insulator phase). However, the measured resistance of the undoped Cd_3_As_2_ film decreases gradually with temperature [58]. By assuming a DC bias of 1 V applied to the device, the photoconductivity current *I*_0_ can be roughly estimated by the measured resistance. Figure 6 shows the extracted CPGE current normalized by the excitation power and photoconductivity current *I*_0_, respectively, as a function of temperature for the doped film of Ni-4% and the undoped one.

As can be seen from Figure 6b, after eliminating the influence of the optical absorptivity and the electron mobility, the temperature variation trend of the normalized CPGE current relative to the photoconductivity current *I*_0_ for both the doped and undoped films is almost the same as that in Figure 6a, implying that the optical absorptivity and electron mobility are not the main factors affecting CPGE current variation with temperature. A previous study has shown that *α_e_* increases slowly with temperature [63]. The temperature variable *τ_p_* is generally more significant than that of *τ*_0_. In addition, our previous work has shown that the photoexcited carrier relaxation time (a few picoseconds) in Cd_3_As_2_ films gradually increases with temperature [59,64]. And the carrier density is known to increase with temperature as well [58,65]. Therefore, under the same excitation pump power, the higher-density carriers will be excited at the elevated temperature, though the electron mobility decreases with increasing temperature. Moreover, it is expected that the enhanced Coulomb screening at higher carrier density would efficiently weaken the electron-electron and electron-phonon scattering [64], thus can slow down the momentum relaxation process with increasing temperature. This factor may be the main reason for the general CPGE current increase with rising temperature in the undoped Cd_3_As_2_ film observed in our experiment, though we cannot quantitatively describe the exact influence of different factors on the temperature variation of CPGE current.

For Ni-doped Cd_3_As_2_ thin films, though its Curie temperature *T*_c_ was determined to be around 45 K [57,58], there is no obvious abnormal change of CPGE near *T*_c_ observed as seen in Figure 6, indicating that magnetism has little influence on CPGE current in the Ni-doped Cd_3_As_2_ films. In the temperature range of 10–75 K, CPGE current gradually increases with temperature, this tendency is consistent with that in the undoped Cd_3_As_2_ film shown in Figure 6. We believe that it has the same reason, that is, the increased carrier density and suppressed momentum relaxation with rising temperature lead to the increase of CPGE current. As temperature rises beyond 75 K, the increased Fermi level could be closer or even higher than the deep-level magnetic impurity state introduced by Ni doping, in addition, the occupation probability of states nearer the Fermi energy is strongly affected by the smeared Fermi surface upon optical excitation [66]. These facts can promote the efficient capture of photo-generated carriers by the deep-level magnetic impurity bands. The trapped electrons are unable to contribute effective photocurrent, resulting in an obvious decrease of CPGE current. As a matter of fact, our previous transient reflection studies could reveal the long decay time of tens of picoseconds associated with the trapped carriers for the Ni-doped films only at temperatures above 100 K [59]. This is, to some extent, in line with the drastic decrease of CPGE above 75 K observed here for the Ni-doped film. When temperature increases up to 200 K, it is seen that the CPGE current increases again when temperature increases. The possible reason for this phenomenon may be that the impurity states near the Fermi level gradually ionize at higher temperatures, and more free carriers can contribute to CPGE current generation. So that the overall CPGE current increases gradually again with temperature, similar to that in the undoped Cd_3_As_2_ film.

## 4. Conclusions

We systematically investigated the CPGE current generation in the epitaxial Ni-doped Cd_3_As_2_ films with different doping concentrations at various temperatures. Our experimental results show that, though the Ni-doped Cd_3_As_2_ films become ferromagnetic below 45 K owing to the magnetic doping, the observed CPGE current in the Ni-doped Cd_3_As_2_ film does not get enhanced by the possible transition from DSM to Weyl semimetal owing to the broken time-reversal symmetry. Instead, we observed two orders of magnitude smaller CPGE current in the Ni-doped Cd_3_As_2_ films than that in the undoped one under the same experimental conditions. The CPGE generation in the Ni-doped Cd_3_As_2_ films was attributed to the reduced structural symmetry by the large epitaxial strain and magnetic doping, the same as the undoped film. Moreover, the CPGE current in the Ni-doped Cd_3_As_2_ films exhibits a complex change when varying temperature, unlike the undoped film which shows a generally increased CPGE current with rising temperature. The greatly reduced CPGE current generation efficiency in the Ni-doped Cd_3_As_2_ films, together with its complex temperature variation, was closely connected with the efficient capture of photo-generated carriers by the deep-level magnetic impurity bands and the enhanced momentum relaxation caused by strong impurity scattering when doped with magnetic elements. An improved magnetic doping technique with reduced or absent magnetic impurity levels is expected to greatly enhance the CPGE current generation efficiency in magnetically doped Cd_3_As_2_ films.

## Figures and Tables

**Figure 1 nanomaterials-13-01979-f001:**
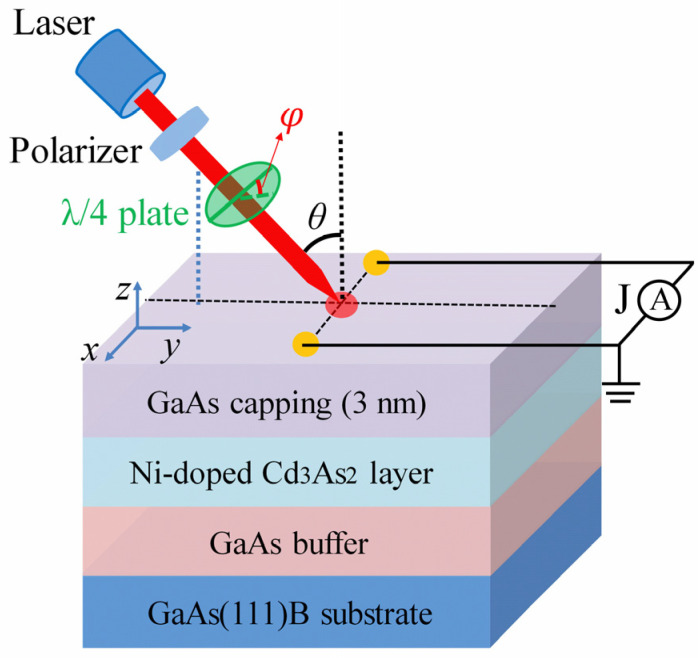
The experimental sketch showing the photo excitation and current measurement. The optical incident angle is *θ*. The *x*-*y* coordinates plane is defined to be parallel to the sample’s surface. The red circle in the center is the laser spot, and the yellow circles along the *x*-axis are the ohmic electrodes. A λ/4 wave plate is utilized to tune the light helicity (by tuning the angle *φ* between the linear polarization direction of the excitation laser and the fast axis of the λ/4 plate). The photocurrent is collected along the *x* direction.

**Figure 2 nanomaterials-13-01979-f002:**
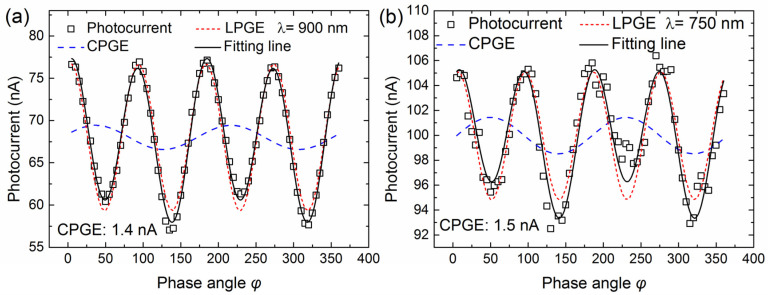
The photocurrent response measured at 10 K by varying phase angle *φ* for the Ni-4% sample. The incident angle is 45° and the excitation power is 14 mW. The excitation wavelength in (**a**,**b**) is 900 nm and 750 nm, respectively. The open squares represent the raw experimental data, and the black lines represent the fitting curve based on Equation (1). The extracted CPGE and LPGE currents are represented by the blue and red dashed lines, respectively.

**Figure 3 nanomaterials-13-01979-f003:**
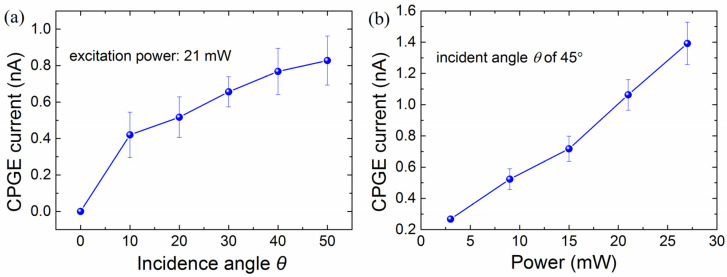
(**a**) The extracted CPGE amplitude at various incident angles *θ* measured with the excitation power of 21 mW. (**b**) The CPGE photocurrent at various excitation power from 3 to 27 mW measured at the incident angle of 45°. The measurements are done at 10 K with an excitation of 900 nm for the Ni-4% sample-B.

**Figure 4 nanomaterials-13-01979-f004:**
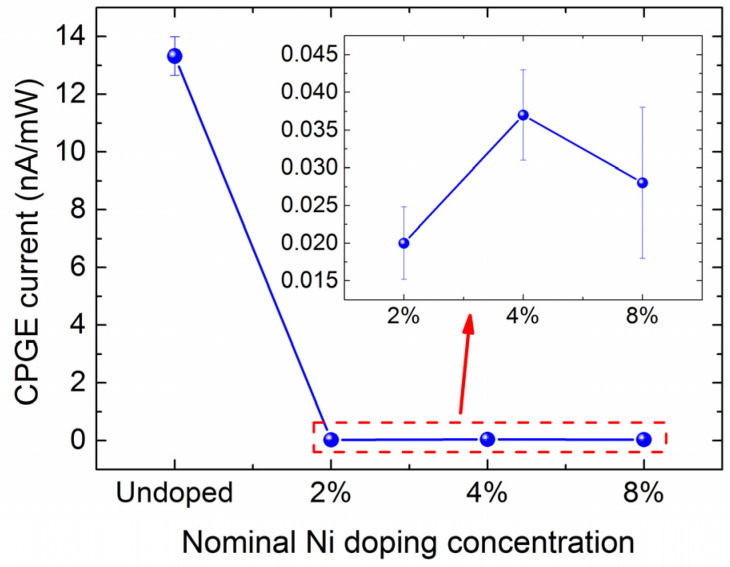
The measured CPGE current normalized with the corresponding photoexcitation power in the Ni-doped Cd_3_As_2_ films with different Ni doping concentrations. The measurements are done at room temperature with excitation of 900 nm at an incident angle of 45°. The inset shows the zoom-in results for the doped films of Ni-2%, Ni-4% and Ni-8%, respectively.

**Figure 5 nanomaterials-13-01979-f005:**
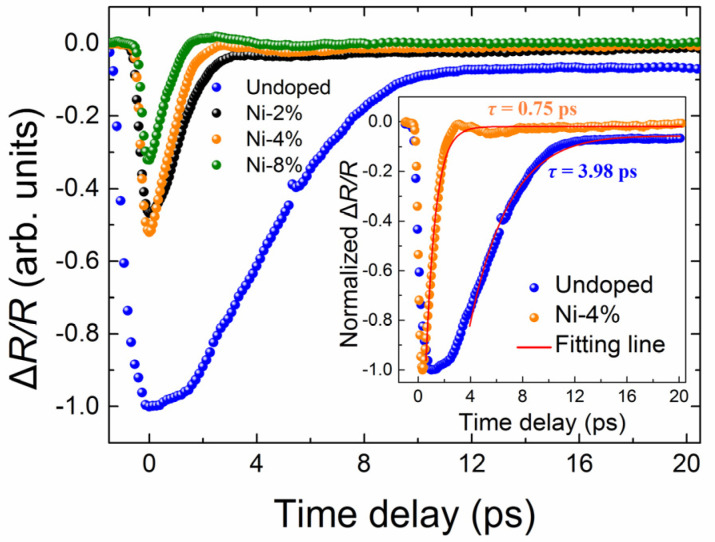
The Δ*R*/*R* responses of Cd_3_As_2_ films with different Ni doping concentrations at 4.5 K, measured with the pump and probe wavelength of 800 nm and 3500 nm, respectively. All the Δ*R*/*R* responses were normalized relative to the minimum value of the undoped sample. The inset shows the fitting results for the undoped and doped (sample B, Ni-4%) samples, here the Δ*R*/*R* responses were normalized relative to their own minimum values of each sample.

**Figure 6 nanomaterials-13-01979-f006:**
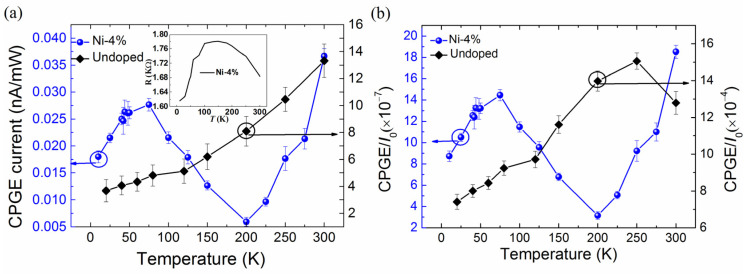
(**a**) The extracted CPGE generation efficiency at various temperatures, obtained by normalization of the measured photocurrent with respect to the corresponding pumping power, for the doped film of Ni-4% (the blue dots) and the undoped one (the black diamond), respectively. The measurements are done at an incident angle of 45° under excitation of 900 nm and 750 nm, respectively. The inset shows the temperature-dependent resistance measured for the doped film of Ni-4%. (**b**) Temperature dependence of CPGE current normalized relative to the photoconductive current *I*_0_ for the doped film of Ni-4% (the blue dots) and the undoped one (the black diamond), respectively.

**Table 1 nanomaterials-13-01979-t001:** The electron density and Hall mobility at 3 K for different samples [58].

Sample	Undoped	Sample-A	Sample-B	Sample-C
Eectron density (cm^−2^)	7.8 × 10^11^	8.0 × 10^11^	1.4 × 10^12^	1.1 × 10^12^
Hall mobility (cm^2^ V^−1^ s^−1^)	710	1000	560	260

## Data Availability

The data supporting the findings of this study are available from the corresponding author upon reasonable request.

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
