# Peer review of "Circular Photogalvanic Current in Ni-Doped Cd3As2 Films Epitaxied on GaAs(111)B Substrate"

_nanomaterials, 2023, doi:10.3390/nano13131979_

Round 1

Reviewer 1 Report

The manuscript reports the circular photogalvanic effect (CPGE) of Ni-doped Cd3As2 films on GaAs (111)B substrate. The Ni doping reduced CPGE current generation efficiency by the increased capture of photo-generated carriers and momentum relaxation. The results in the manuscript reasonably support the role of doped-Ni on CPGE, but the following comments can be considered.

1) In the introduction, the authors quoted the references relating to Cr- and Mn-doped Cd3As2 on CPGE. Certainly, the study of another 3d transitional metal element can be informative to understand CPGE of Cd3As2, but the motivation to emphasize Ni doping is unclear.

2) The authors claimed that the CPGE current generation in the Ni-doped Cd3As2 can be attributed to the lattice strain. The quantitative values of the increased lattice strain by 2%, 4%, and 8% Ni doping can be helpful to readers.

3) Lines 267-268: Are there references to affecting a phase transition by Ni or other 3d-TM doping? How to determine a phase transition? Is it based on the resistance change behavior with temperature?

4) In the conclusion section, the influence of Ni-doping on Cd3As2 films was summarized. For the application concerns, Ni-doping appears to present adverse effects. The authors may suggest the direction of a doping study for optoelectronic applications.

Reviewer 2 Report

The manuscript presents investigation of the  circular photogalvanic effect  for the Ni-doped 13 Cd3As2 films. Authors conclude that the effect originates from the structural symmetry breaking induced by the  lattice strain and magnetic doping.

In my opinion, the most striking result is that the effect  in the Ni-doped Cd3As2 films was approximately two  orders of magnitude smaller than that in the undoped one under the same experimental conditions. This is just opposite to the expected behavior: no circular photogalvanic effect should be observed in centrosymmetric Cd3As2, while the symmetry should be destroyed by doping with magnetic impurities.  On the other hand, magnetism was found to have  little influence on CPGE current in the  Ni-doped Cd3As2 films.

This discrepancy is ascriped to the structural effects in the present manuscript. This may be correct, but no structural investigations can be found in the manuscript. Thus, the claim of the paper can not be confirmed by experimental results, which is obviously a problem.

Reviewer 3 Report

Reviewers report: The manuscript “Circular photogalvanic current in Ni-doped Cd3As2 films epitaxied on GaAs (111)B substrate“ devoted to study  circular photogalvanic effect in Cd3As2 films grown on GaAs (111) B substrate by molecular beam epitaxy. It was found that in Ni-doped Cd3As2 films circular photogalvanic current significantly smaller than in undoped one. The greatly reduced circular photogalvanic current in Ni-doped Cd3As2 films was explained by efficient capture of photo generated curriers by the deep-level magnetic impurity bands and the enhanced momentum relaxation caused by strong impurity scattering. The experimental measurements were accomplished with great accuracy.

The manuscript represents a significant new contribution to investigation of properties of epitaxially grown Ni-doped Cd3As2 films and can be accepted for publishing.

Round 2

Reviewer 2 Report

The authors answer my questions. Most important, that the conclusion is confirmed by direct investigation of the sample structure. Thus, the interpretation has a good chance to be correct.